# Understanding the mechanism of catalytic fast pyrolysis by unveiling reactive intermediates in heterogeneous catalysis

Patrick Hemberger[1], Victoria B.F. Custodis[2,3], Andras Bodi[1], Thomas Gerber[4] & Jeroen A. van Bokhoven[2,3]

Catalytic fast pyrolysis is a promising way to convert lignin into fine chemicals and fuels, but current approaches lack selectivity and yield unsatisfactory conversion. Understanding the pyrolysis reaction mechanism at the molecular level may help to make this sustainable process more economic. Reactive intermediates are responsible for product branching and hold the key to unveiling these mechanisms, but are notoriously difficult to detect isomer-selectively. Here, we investigate the catalytic pyrolysis of guaiacol, a lignin model compound, using photoelectron photoion coincidence spectroscopy with synchrotron radiation, which allows for isomer-selective detection of reactive intermediates. In combination with ambient pressure pyrolysis, we identify fulvenone as the central reactive intermediate, generated by catalytic demethylation to catechol and subsequent dehydration. The fulvenone ketene is responsible for the phenol formation. This technique may open unique opportunities for isomer-resolved probing in catalysis, and holds the potential for achieving a mechanistic understanding of complex, real-life catalytic processes.

[1] Laboratory for Femtochemistry and Synchrotron Radiation, Paul Scherrer Institute, CH-5232 Villigen-PSI, Switzerland. [2] Department of Chemistry and Applied Biosciences, Institute for Chemical and Bioengineering, ETH Zurich, HCI E 127, Vladimir-Prelog-Weg 1, 8093 Zurich, Switzerland. [3] Laboratory for Catalysis and Sustainable Chemistry, Paul Scherrer Institute, OSUA/201, CH-5232 Villigen-PSI, Switzerland. [4] Molecular Dynamics Group, Paul Scherrer Institute, CH-5232 Villigen-PSI, Switzerland. Correspondence and requests for materials should be addressed to P.H. (email: patrick.hemberger@psi.ch) or to J.A.v.B. (email: jeroen.vanbokhoven@chem.ethz.ch).

Unravelling reaction mechanisms is often notoriously difficult to achieve. New details on the $S_N2$ mechanism have emerged thanks to advances in computational chemistry[1,2], and crucial oxidation intermediates have been detected in combustion only recently[3]. In heterogeneous catalysis, intermediates are formed from activated reactants, and govern product selectivity and conversion. As the reactive intermediates are short-lived, they are even more challenging to identify experimentally. Surface sensitive methods often suffer from spectral congestion when numerous isomers and compound classes react in a complex network of reactions[4–6]. Among gas-phase approaches, photoionization mass spectrometry stands out as a universal, sensitive, selective and time-resolved detection tool[7]. However, it only provides limited spectral information, often complicating the assignment of isomers. This challenge can be overcome by coincident detection of the photoelectron and the mass spectrum. Imaging Photoelectron Photoion Coincidence Spectroscopy (iPEPICO) offers superior isomer selectivity[8], for instance in determining the unimolecular decomposition mechanism of fire retardants[9], combustion relevant radicals[10] or even when studying flames directly[11,12].

Can iPEPICO also help us to elucidate catalytic reaction mechanisms based on the intermediates desorbed from the catalyst surface? On the basis of the results of the zeolite-catalysed pyrolysis of guaiacol presented herein, we believe it can. With the help of such mechanisms, we can adapt and develop catalytic processes to maximize conversion and product selectivity, turning, for example, biomass into a sustainable source of renewable fuels and value-added chemicals. To ensure it is also an economically viable one, we must understand how to convert waste biomass to high-value products[13]. Lignin is, after cellulose and hemicellulose, the third major component in biomass and its backbone is highly aromatic, containing phenolic, alkoxy and aryloxy subunits. Because of the high bond energies, the cross-linked macromolecular structure, and hydrophobic nature of lignin, it must be processed to remove oxygen and obtain small aromatic hydrocarbons and olefins[14]. Catalytic fast pyrolysis (CFP) is probably the most promising approach[15,16], in which the lignin macromolecule is depolymerized into phenolic subunits, such as guaiacol, which are then converted into aromatics in zeolite-catalysed pyrolysis[17,18]. Guaiacol also contains typical oxygen functionalities in lignin and thus serves as model compound[19,20]. Especially when empirical 'cook-and-look' strategies fail, understanding the reaction mechanism on a molecular level is essential to optimize the catalysts' performance and to maximize conversion and selectivity in catalytic processes. Because the identity of the intermediates is unknown experimentally, the deoxygenation mechanism in catalytic pyrolysis remains poorly understood. Hydrodeoxygenation mechanisms on platinum, nickel or nanoparticle catalysts have been investigated computationally[21] and experimentally[19,20,22], identifying catechol as the main intermediate from guaiacol demethylation. However, the mechanism to yield phenol from catechol could only be tentatively explained by a dehydration mechanism on the alumina support, since no reaction pathway was found yielding phenol[21]. Alternation of reaction conditions and reactants have not led to a deeper understanding of the mechanism either[23–25]. We set out to identify the missing reactive intermediates using a temporal analysis of products-type (TAP)[26] pyrolysis reactor coupled with our imaging PEPICO spectrometer (py-iPEPICO) at the VUV beamline of the Swiss Light Source. This combination is unique since it brings together a catalytic reactor, which works in the low-density regime ( < 1 mbar) allowing for the detection of desorbed reactive intermediates using PEPICO, which also provides superior isomer selectivity and fragment free soft ionization[27,28]. Reactive intermediates desorbed from the catalyst surface are quenched rapidly at atmospheric pressure, which means the experiment must take place at low pressures and in the presence of an inert buffer gas. This induces a pressure gap between realistic and experimental catalytic pyrolysis conditions. Therefore, the py-iPEPICO measurements were complemented by ambient pressure batch-type pyrolysis gas chromatography mass spectrometry (py-GC/MS). Reactive intermediates are identified isomer-selectively with the help of the photoelectron spectrum, recorded in coincidence for each $m/z$ peak. The observation of reactive and stable intermediates allows us to propose a catalytic pyrolysis mechanism. Moreover, the concepts and techniques are generally applicable in heterogeneous catalysis.

## Results

**Bridging the pressure gap.** To suppress gas-phase chemistry, guaiacol was diluted in argon and pulsed over a H-USY zeolite catalyst coating in a quartz glass TAP-type reactor held at a constant temperature between 400 and 500 °C. This set-up enables tracing products and reactive intermediates desorbed from the surface, due to the abundance of an inert buffer gas, the high dilution of the reactants and low average pressure in the reactor (Supplementary Methods and Supplementary Fig. 1 for a detailed description of the set-up). The desorbed species enter the ionization region of the iPEPICO endstation with the gas flow and are photoionized by monochromatic synchrotron radiation. The photoelectrons are velocity map imaged, and the photoions are detected in delayed coincidence according to their time of flight, which allows us to record the mass spectrum as well as the photoelectron spectrum belonging to each $m/z$ channel[9,10,29–31].

Figure 1 shows a mass spectrum taken at 10.5 eV photon energy, which can be compared to the selectivity plot of the py-GC/MS set-up (lower trace), obtained in batch-wise operation at ambient pressure. The py-GC/MS enables detection of final products in absolute yields, resulting from a combination of gas phase as well as surface reactions. Phenol, cresols, catechol and xylenols appear in both spectra at $m/z = 94$, 108, 110 and 122, respectively, although their relative intensities differ. First, doubly and triply methylated products, for example, at $m/z = 122$ and 136 are more abundant in the py-GC/MS spectrum, which is a consequence of lower reactant densities on the surface in the py-iPEPICO approach. Together with the increased abundance of catechol at $m/z = 110$, which is the immediate guaiacol demethylation product, this hints at guaiacol being the methyl source. Second, light products and five-membered ring intermediates, for example, at $m/z = 66$ and 80 (Fig. 1), are completely absent in the py-GC/MS spectrum. These species may evade detection at ambient pressure because of the longer detection time scale, gas-phase reactions or because they simply do not survive the GC interface between the reactor and the MS. Together with the heavier desorption products, their isomer-selective assignment may hold the key to the guaiacol conversion mechanism.

**Species identification.** The threshold photoionization matrix (Fig. 2) shows the near-zero kinetic energy electron signal intensity as a function of photon energy and photoion $m/z$ ratio. Horizontal cuts correspond to threshold photoionization mass spectra at the given photon energies, and vertical cuts yield photoion mass-selected threshold photoelectron spectra (ms-TPES) for the selected $m/z$ ratios. The vibrational fine structure of the ms-TPES is an isomer-specific fingerprint, which can be used to assign the peaks in the reaction mixture to a single or multiple isomer(s) with the help of calculated Franck–Condon

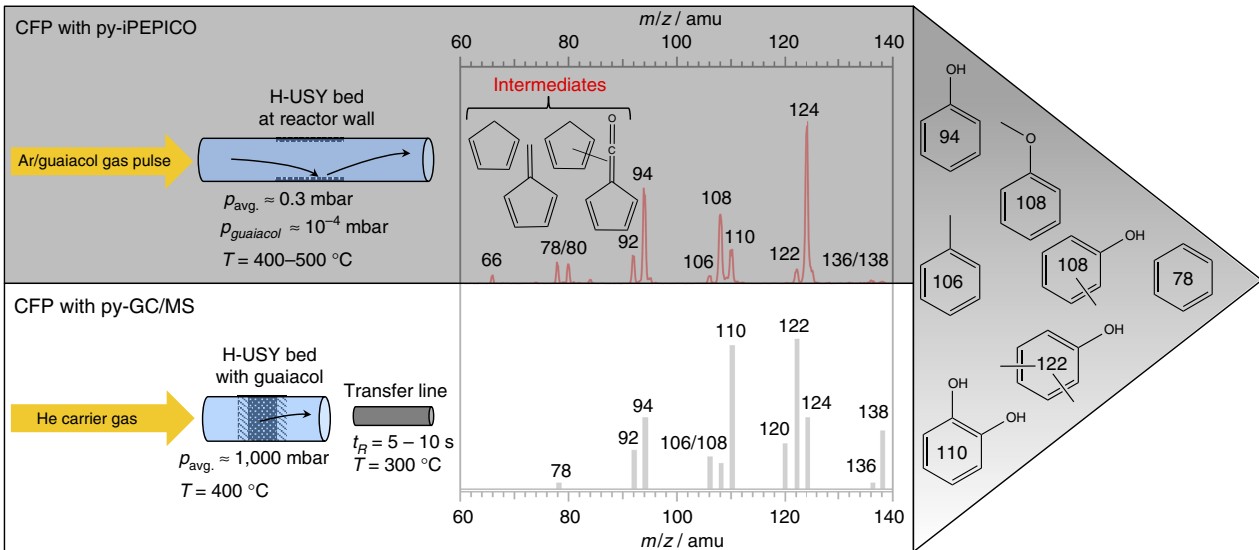

**Figure 1 | Comparison of the two experimental set-ups used in this study.** The py-iPEPICO (upper trace) set-up provides insights in the catalytic reaction and enables isomer-specific detection of reactive and stable intermediates. On the other hand, the py-GC/MS (lower trace) set-up is limited by the long residence time ($t_R$) from the transfer line to the detector, but provides more realistic operating conditions.

(FC) factors[9,10,30–32]. As also seen in the ambient pressure experiments, the major products are phenol, anisole and the three cresol isomers, as well as catechol at $m/z = 94$, 108 and 110, respectively (Supplementary Fig. 2). The ms-TPES for the most relevant species at $m/z = 66$, 78, 80 and 92, partially unique to the py-iPEPICO experiment, are shown individually in the insets Fig. 2a–d. The $m/z = 66$ peak, $C_5H_6$, is readily identified as cyclopentadiene ($c$-$C_5H_6$) with the help of FC simulations. The onset of the $m/z = 78$, $C_6H_6$, ms-TPES, below the benzene ionization energy of $IE = 9.244$ eV (ref. 33), suggests another isomer at play. FC calculations identify fulvene ($c$-$C_5H_4$=$CH_2$) as the dominant contributor. In py-GC/MS experiments, fulvene converts into benzene, escaping detection (*vide infra*).

Although the antiaromatic cyclopentadienone ($c$-$C_5H_4$=O, $m/z = 80$) is a thermal decomposition product of guaiacol[34,35], the $m/z = 80$ ms-TPES exhibits vibrational structure characteristic of three methyl-cyclopentadiene ($c$-$C_5H_5$–$CH_3$) isomers only[29]. The $C_6H_4O$ signal at $m/z = 92$ can be identified as fulvenone (cyclopenta-2,4-dien-1-ylidenemethanone, $c$-$C_5H_4$=CO), based on the strong transition at 8.25 eV. Possible toluene contributions cannot be ruled out at higher photon energies ($IE = 8.828$ eV), but are blended into the strong excited ion state bands of fulvenone[36]. Finally, methyl-fulvenones ($c$-$C_5H_4(CH_3)$=CO) were assigned to $m/z = 106$ (Supplementary Fig. 2). At ambient pressure, fulvenones are quite reactive with respect to thermal decomposition, dimerization or hydrolysis, which probably completes within nanoseconds[37], and only toluene and xylenes could be observed in mass channels 92 and 106 in py-GC/MS, respectively. Carbon monoxide, carbon dioxide and ethenone have also been detected using py-iPEPICO.

In addition to the isomer information, we identified products and intermediates as function of the time-on-stream (TOS) using the py-iPEPICO set-up. These data (Supplementary Fig. 3 and Supplementary Table 1), give information on the catalyst deactivation as well as the appearance of intermediates and products, and the desorption processes.

**Reaction mechanism.** The stoichiometry of the products and intermediates helps us to identify the two major reaction steps

that the conversion mechanism must account for. First, deoxygenation, the loss of one or two oxygen atoms, is responsible for the formation of phenol, cyclopentadiene, fulvenone and fulvene (Fig. 3, red arrows).

Deoxygenation is driven by the stability of the leaving groups: water, carbon monoxide and carbon dioxide. Second, it is evident in the py-iPEPICO spectra, and even more so in the py-GC/MS results that a methyl group pool is distributed among the final products and transmethylation is responsible for the formation of the cresols, methyl-cyclopentadienes and methyl-fulvenones (Fig. 3, blue arrows). This methyl pool can be sourced to a loose methyl group in guaiacol (*vide infra*), the removal of which leaves behind catechol while methylating other intermediates. In view of the product stream species identified based on their ms-TPES, a reaction mechanism can now be constructed and is shown in Fig. 3. The mechanism explains the formation of the reactive species observed here for the first time as well as that of the stable conversion products and intermediates. Three test experiments have been used to validate our mechanism: the methyl group in guaiacol **1** has been [13]C-labelled at the methoxy group and traced, the Brønsted acid sites were removed from the catalyst, and catechol **2** was converted instead of guaiacol to suppress the transmethylation steps.

**Tracking the methyl group.** The methoxy functional group of guaiacol has been [13]C-labelled to follow its fate in the conversion. Catechol **2**, fulvenone **3**, phenol **6** and cyclopentadiene **9** remain non-labelled to more than 90%, while methyl-cyclopentadienes **11**, fulvene **12**, methyl-fulvenones **13** and cresols **14** shift by one $m/z$ unit, indicating methylation with [13]CH$_3$ (Fig. 4a,b). This confirms that the methoxy group of guaiacol acts as methylating agent in a Brønsted-acid catalysed fashion on the surface (indicated as blue arrows in Fig. 3), and unlabelled catechol **2** is produced as the demethylation product.

Furthermore, the ring in phenol **6**, catechol **2** as well as the cresols **14** remain unlabelled, which proves the integrity of the $C_6$ aromatic moiety without its destruction and reconstruction. The original $m/z = 78$ signal is split in two upon [13]C-labelling (Fig. 4a,b), and the ms-TPES reveals that the 78 amu signal is mostly non-labelled benzene **7**, while the 79 amu signal is almost

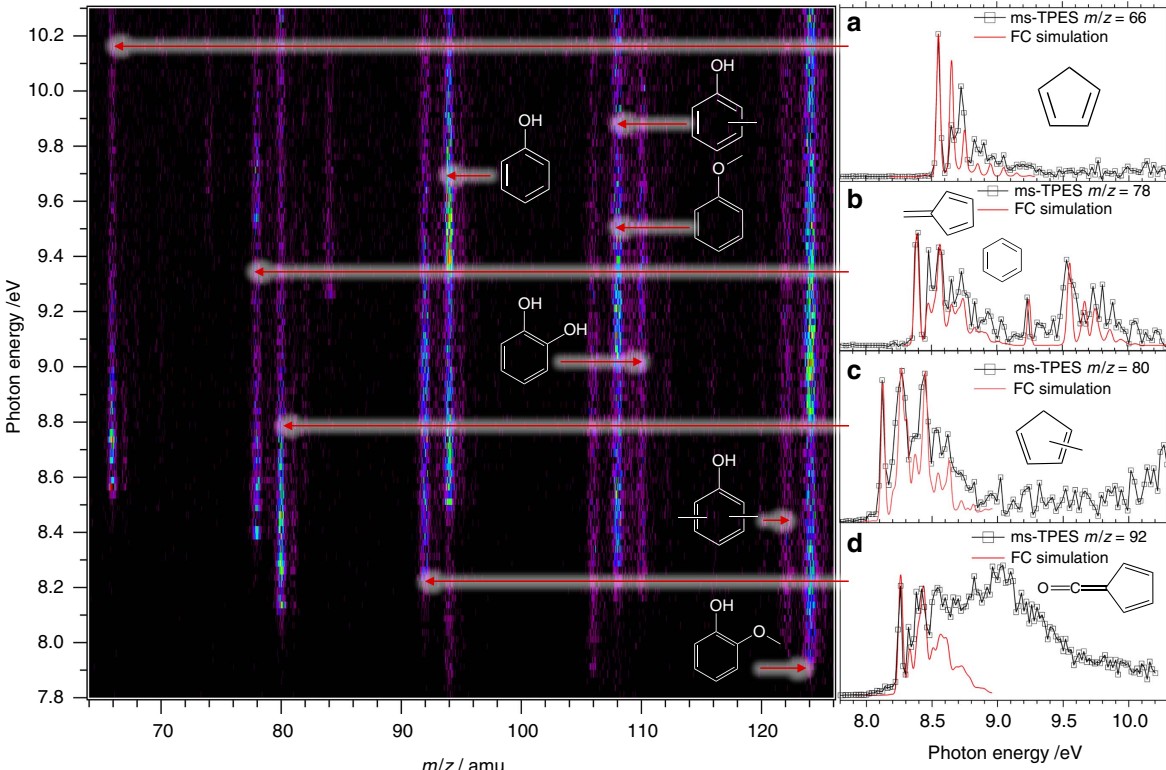

**Figure 2 | Threshold photoionization matrix of the detected products and intermediates upon catalytic pyroylsis.** Insets (**a**–**d**) show ms-TPE spectra for selected *m/z* channels assigned isomer-specifically with the help of Franck–Condon simulations as cyclopentadiene (*m/z* = 66), fulvene (*m/z* = 78), methyl-cyclopentadienes (*m/z* = 80) and 6-fulvenone (*m/z* = 92).

exclusively [13]C-fulvene **12**. This warrants the two paths shown in the reaction mechanism: labelled methyl-cyclopentadiene dehydrogenates to yield fulvene (**11→12**), whereas unlabelled phenol is deoxygenated to yield benzene (**6→7**). In the py-GC/MS experiment with labelled guaiacol, both labelled and unlabelled benzene has been detected, indicating complete isomerization in the batch-type reactor, in the transfer line or in the GC/MS to the most stable $C_6H_6$ isomer benzene (Supplementary Fig. 4), and confirming that fulvene may not be detected in the py-GC/MS set-up due to gas-phase or surface-catalysed rearrangement. The presence of only singly labelled benzene and fulvene in both experiments proves that five- and the six-membered ring species are strongly connected through a transmethylation reaction and are not formed by complete reconstruction of the ring.

Phenol and anisole are assumed to be final products in the reaction mechanism, but they could decarbonylate and yield cyclopentadiene[38] and benzene[15], respectively. Because of the small reactivity of pure phenol over H-USY and the missing labelled products ([13]CO) from labelled anisole, deoxygenation of phenol and anisole could also be excluded over H-USY[39,40].

**Removal of Brønsted acid sites**. When guaiacol is thermally decomposed unimolecularly in a microreactor in a noble gas stream, the main product is cyclopentadienone after demethylation, decarbonylation and hydrogen atom loss[34,35]:

$$C_6H_4(OH)(OCH_3) \rightarrow \cdot CH_3 + C_6H_4(OH)(O\cdot)$$
$$\rightarrow CO + \cdot CH_3 + \cdot H + c-C_5H_4{=}O \quad (1)$$

Brønsted acid sites are supposed to initiate both deoxygenation and transalkylation reactions on the H-USY, which explains why neither thermal decomposition intermediates nor cyclopentadienone were observed in the py-iPEPICO experiment. This was further confirmed by exchanging Brønsted acid sites with sodium ions in USY, which yielded both methyl radicals as well as cyclopentadienone at a reactor temperature of 500 °C (Supplementary Fig. 5).

Methyl radicals are indeed the only radical species observed in the py-iPEPICO experiment and are only visible in the absence of acid sites on the catalyst[41]. Even over Na-USY, only cyclopentadienon ($c$-$C_5H_4{=}O$) was observed as the decomposition product of the hydroxyphenoxy counter fragment after $H_3C$–O bond cleavage at $m/z = 109$ (eq. 1)[34]. Similar to the methyl radical, other radical intermediates are expected to survive the few wall collisions within the reactor in the few millisecond transfer time to detection. Several indications point to a strong binding of the radicals to the catalyst surface or a rapid quenching before desorption: First, by using electron paramagnetic resonance (EPR) spectroscopy, Bährle *et al.*[42,43] showed a radical concentration enhancing effect during catalytic pyrolysis of lignin. This is indicative for a radical strongly bound to the zeolite surface by chemisorption. Second, a strong adsorption of guaiacol on the surface is observed in the py-iPEPICO data, which is mirrored by an almost two orders of magnitude drop in the ion signal in the presence of a catalyst (Supplementary Note 1). Using Na-USY in our py-iPEPICO set-up results in the decomposition of guaiacol to unimolecular decompositions products, accompanied by few transmethylation reactions as summarized in Supplementary Fig. 5. Similar observations were made utilizing the py-GC/MS set-up together with Na-USY proving again the low reactivity of the catalyst in the absence of Brønsted acid sites. Some transmethylation products could be assigned along with phenol, hydroxybenzaldehyde and diphenylether, which are the result of bimolecular

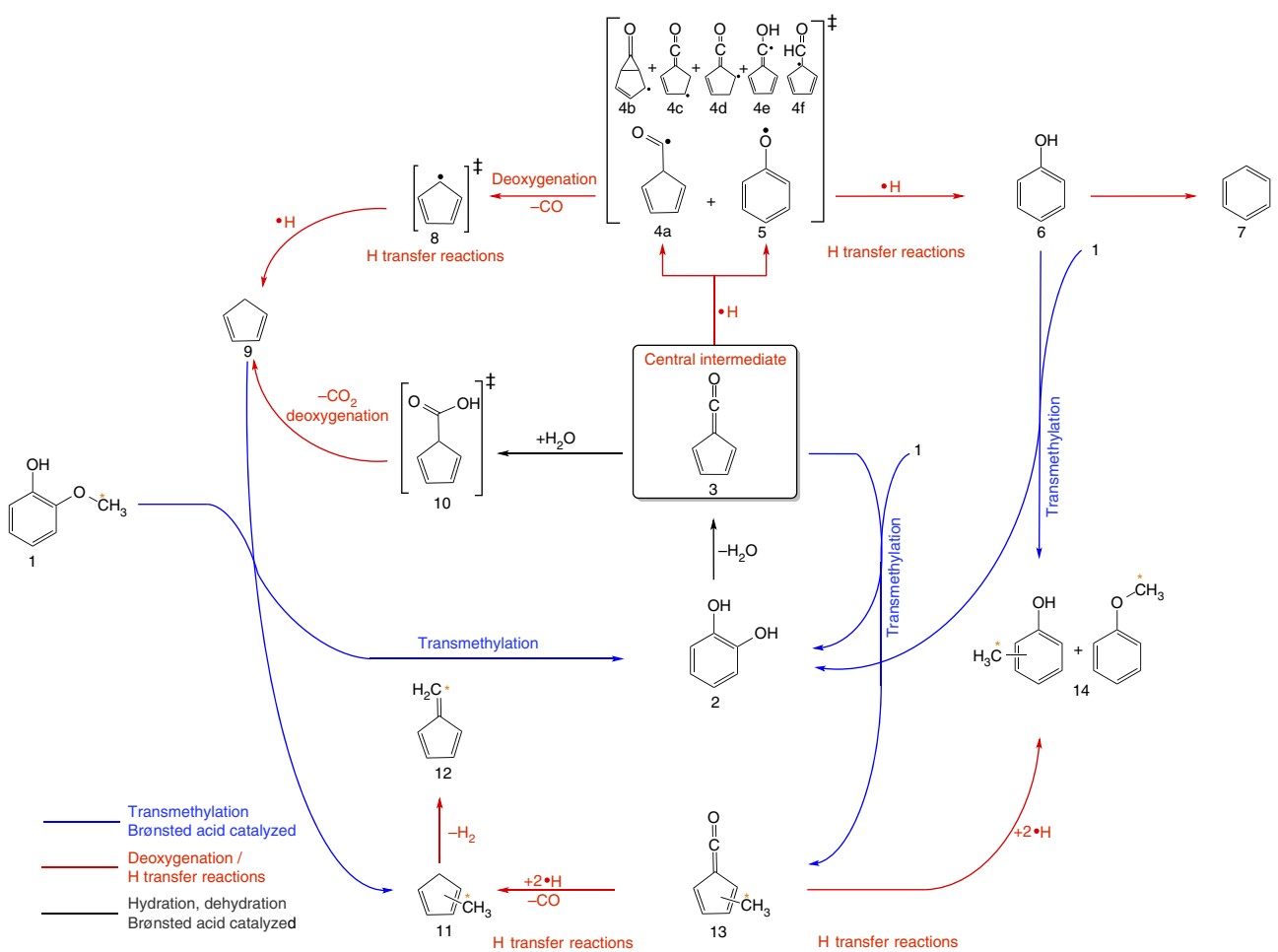

**Figure 3 | Overview of the whole reaction mechanism of H-USY zeolite-catalysed pyrolysis of guaiacol.** The colour-coding of the reaction arrows denotes the type of reaction leading to the products. Products appearing [13]C-labelled are marked with an asterisk. Unobserved, but inferred species are labelled with a double dagger.

chemistry, probably happening in the gas phase, because they are also observed in absence of a catalyst[34].

**Suppressing transmethylation.** Catechol **2** is the primary transmethylation product of guaiacol **1**, which, in contrast to phenol, is assumed to react further on the catalyst surface. Indeed, catechol is inert even at 600 °C without catalyst, but it decomposes already at 400 °C in the presence of H-USY to fulvenone **3**, phenol **6** and cyclopentadiene **9**.

Using catechol as reactant depletes the methyl pool and inhibits transmethylation, which is evidenced by the conspicuous lack of methylated products in Fig. 4c. Fulvenone is facilely formed by dehydration, but is it also an intermediate to phenol and cyclopentadiene? Room temperature CBS-QB3-calculated reaction enthalpies are as follows[44]:

$$C_6H_4(OH)_2 \rightarrow H_2O + c-C_5H_4=C=O \qquad \Delta H_r = 143.7\,\text{kJ mol}^{-1} \tag{2}$$

$$C_6H_4(OH)_2 \rightarrow CO_2 + c-C_5H_6 \qquad \Delta H_r = 17.6\,\text{kJ mol}^{-1} \tag{3}$$

Water elimination (eq. 2) is endothermic in the gas phase, but could be promoted by hydrogen bonding on the catalyst surface and needs to pass only one transition state[45,46]. Decarboxylation (eq. 3) is isenthalpic, but gas-phase calculations suggest that it

also proceeds by the fulvenone ketene as intermediate. Fulvenone ketene is then re-hydrated to yield cyclopentadienyl acid (eq. 4), which subsequently decarboxylates:

$$H_2O + c-C_5H_4=C=O \rightarrow c-C_5H_5-COOH$$
$$\rightarrow CO_2 + c-C_5H_6 \quad \Delta H_r = -126.1\,\text{kJ mol}^{-1}. \tag{4}$$

Alternatively, fulvenone may pick up a hydrogen atom, and subsequently lose carbon monoxide to yield cyclopentadiene (see below). On the one hand, the fulvenone, carbon monoxide, carbon dioxide and cyclopentadiene formation is explained in the catalytic decomposition of catechol. Phenol, on the other hand, still remains unaccounted for.

Despite deactivating the catalyst, coke and carbonaceous deposits can also promote hydride and proton transfer from condensed polyaromatics to other intermediates bound on the surface[47,48]. Tetralin, tetrahydronaphthalene, is a hydrogen donor because of the added aromatic stabilization of naphthalene. If tetralin is co-fed to the pyrolysis, it increases the available hydrogen supply and can thus reduce coke formation as it competes with PAH dehydrogenation and condensation[49]. We suggest that hydrogen transfer from highly condensed polyaromatics in coke plays a major role in phenol and cyclopentadiene formation (indicated as red arrows in Fig. 3). Indeed, highly condensed polyaromatics were found using [13]C solid state NMR spectroscopy, which can provide the required

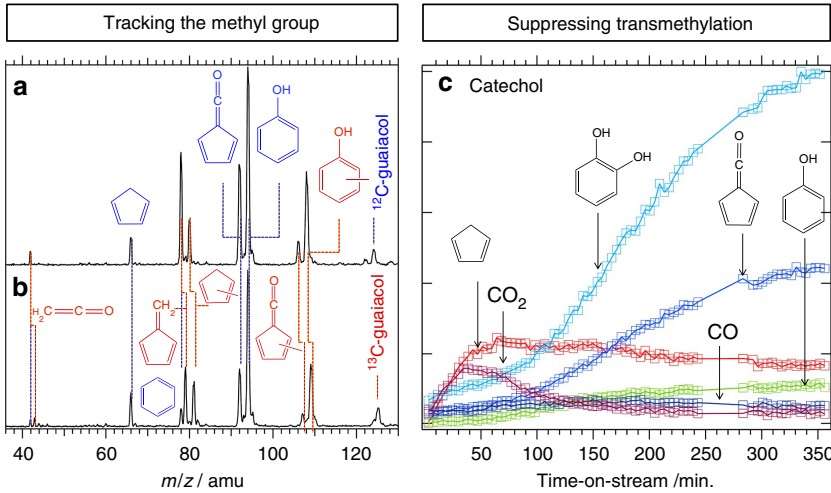

**Figure 4 | Tracking the methyl group and suppressing transmethylation.** Mass spectra show a unit mass shift for fulvene, methyl-cyclopentadiene, the cresols and methyl-fulvenone (labelled red, **a,b**) upon using $^{13}$C-guaiacol. Structures marked in blue are not subject to labelling. Suppressing transmethylation (**c**): Time-on-stream (TOS) curve taken at 14.1 eV in the py-iPEPICO set-up. Pyrolytic catalysis of catechol over H-USY leads to formation of cyclopentadiene ($m/z = 66$), fulvenone ($m/z = 72$) and phenol ($m/z = 94$). The entrance channel to this reaction is the dehydration to yield fulvenone. Apart from these species CO and $CO_2$ are observed as well.

hydrogen atoms[50]. Fulvenone may readily react with a hydrogen from highly condenses polyaromatics on the surface, leading to $C_6H_5O(H)$ species in an exothermic reaction, thanks to the gained aromaticity of tetralin- and dialin-like structures yielding aromatic naphthalenes:

$$c-C_5H_4{=}C{=}O + C_{10}H_{10} \rightarrow C_6H_5-OH + C_{10}H_8$$
$$\Delta H_r = -172.2 \, \text{kJ mol}^{-1} \quad (5)$$

Although this process of hydrogen addition to the fulvenone species may proceed in a concerted fashion, the radical intermediates after formal hydrogen atom addition may explain the branching into cyclopentadiene and phenol: Fig. 3 shows the five possible hydrogen addition sites (**4a–f**). They may interconvert rapidly, and two equilibrium structures (Supplementary Fig. 6), carbonyl-cyclopentadienyl (**4a**) and the phenoxy radical (**5**) intermediates, may explain phenol and cyclopentadiene formation. The connection of phenoxy and cyclopentadienyl radicals was already established in gas-phase experiments and may also be extrapolated to the reactivity on the catalyst surface, justifying a reaction mechanism dominated by hydrogen transfer[30]. Although the phenoxy radical **5** is more stable than the cyclopentadienyl radical **8** (Supplementary Fig. 6), the formation of the latter is irreversible because CO can quickly be removed from the reaction. Thus, the product branching ratio will be determined by the CO removal and the phenoxy thermalization or H-addition rate, whichever faster.

In addition, evidence for radical species adsorbed on the catalyst surface was found by Bährle *et al*,[43] who used EPR spectroscopy to monitor these intermediates upon CFP of lignin on zeolites. Probably due to strong chemisorption on the surface, cyclopentadienyl and phenoxy species will not desorb in our case and are supposed to be rather immobile.

The cyclopentadienyl (**8**) and phenoxy radical (**5**) products readily undergo further hydrogen atom addition, yielding phenol (**5→6**) and cyclopentadiene (**8→9**) as final products, thanks to the high exothermicity of this step.

This may explain why these intermediates are either rapidly quenched on the surface by hydrogen addition and thus evade detection (desorption rates < reaction rate), or stay on the surface due to strong chemisorption and contribute also to the formation

of highly condensed polyaromatics again delivering hydrogen. The formal addition of the second hydrogen atom to the radical intermediate is favoured, because it converts two radicals into two closed-shell species, while forming a double bond in the condensed aromatic ring systems, which are part of char formed on the surface. Phenol and cyclopentadiene are weakly bound to the catalyst and may desorb.

Fulvenone ketene hydrolysis (**3→10→9**) yields $c$-$C_5H_6$ and $CO_2$, and can also compete with hydrogen addition. At the beginning of the reaction, the coke concentration on the catalyst is low and hydrogen addition is less likely. There is indeed a correlation between the carbon dioxide signal and the $c$-$C_5H_6$ signal in the TOS curve of catechol (Fig. 4c). Both increase at first, but the carbon dioxide signal decreases faster than the $c$-$C_5H_6$ signal above 60 min TOS. This is consistent with decarboxylation (**3→10**) being preferred at small coke concentrations. As coke builds up, and leads to an increase of highly condensed polyaromatics on the surface, hydrogenation takes over, generating phenol (**3→5→6**) and also cyclopentadiene (**3→4a→8→9**). Addition of hydrogen from highly condensed polyaromatics yields phenol and cyclopentadiene in exothermic reactions.

In the py-GC/MS set-up, guaiacol is mixed with the catalyst and quickly heated to 400 °C. The local concentration of species on the surface and in the gas phase is much higher, leading to a rapid quenching by bimolecular processes as opposed to the py-iPEPICO set-up, in which there is a flow of diluted reactant. The intermediately formed cyclopentadienyl radicals, for example, can thus quickly dimerize to yield dialin and naphthalene (highly condensed polyaromatics) at ambient pressure:

$$2c-C_5H_5 \rightarrow C_{10}H_{10} \rightarrow C_{10}H_8 + H_2 \quad (6)$$

## Discussion

We have established a reaction mechanism for the catalytic pyrolysis of the lignin monomer guaiacol, based on the isomer-specific assignment of the desorbed products and intermediates from the catalyst surface. The stable products agree well with those in an ambient pressure reactor, detected using GC/MS. The low-pressure environment suppresses gas-phase bimolecular reactions and allows for the identification

of the desorbed intermediates based on their ms-TPES and Franck–Condon photoelectron spectrum simulations. In short, in the presence of Brønsted acid sites on H-USY, guaiacol acts as methylating agent for its own conversion intermediates, yielding catechol as intermediate product. Catechol is dehydrated to form the highly reactive fulvenone ketene, the newly identified central species in the reaction mechanism. Radical driven hydrogenation of fulvenone is responsible for the branching into cyclopentadiene and phenol, thanks to the pool of highly condensed polyaromatics, established when coke is deposited in the zeolite pores. Fulvenone may also be hydrated, and can yield cyclopentadiene by releasing carbon dioxide. Finally, cresols, methyl-cyclopentadiene and fulvene are the result of transmethylation by guaiacol. Five-membered ring species are not detected at ambient pressure, because of consecutive and bimolecular reactions. Fulvene, for example, may rearrange during transfer to the GC/MS in the gas phase and reactive hydrocarbons may be reformed by readsorption onto the catalyst, or in bimolecular reactions in the gas phase, and thus evade detection. The methyl radical was observed when using Na-USY, corroborating our radical detection capability and confirming that other radical intermediates may strongly be chemisorbed on the surface.

The mechanism was confirmed by control experiments: tracking the methoxy carbon in guaiacol by $^{13}C$-labelling, by substituting the Brønsted acid sites to confirm their role in transmethylation, and by turning off the methyl source for transmethylation by pyrolysing catechol.

Transmethylation, hydration, dehydration reactions as well as the hydrogen transfer happen surface-catalysed, as confirmed by the presence of the same products in both set-ups. Furthermore, typical gas-phase reaction products of guaiacol pyrolysis such as diphenylether and hydroxybenzaldehyde were absent in the presence of the H-USY catalyst. Reactive intermediates, such as methyl radicals and the fulvenone ketene could only be detected in the py-iPEPICO set-up speaking for their rapid quenching in the ambient pressure py-GC/MS experiments in the gas phase.

Fulvenone is, thus, the missing reactive intermediate in the deoxygenation of catechol. We infer that this intermediate may also play a crucial role in deoxygenation of lignin in CFP. Decreasing the residence times of intermediates and reactants on the catalyst or in the reactor would lead to an increased lifetime of these species, which may be a way to fine tune the product distributions. In fact, in the py-iPEPICO experiments we observe less multiply methylated species (xylenols or trimethyl phenols) compared to the batch-type py-GC/MS set-up.

More broadly, this study illustrates how the isomer-selective detection of weakly bound intermediates can help fill in the missing gaps in heterogeneous catalytic reaction mechanisms, and how py-iPEPICO uniquely complements surface sensitive analysis techniques by detecting reactive molecules in the gas phase. Insights into catalytic reaction mechanisms and the identification of central reactive intermediates enable the design and development of novel and better catalysts and catalytic processes; for instance, by varying the composition of reactive sites and pore sizes, to fully convert a key precursor into the desired product by passing through a selectivity-determining intermediate. The concepts and technologies used in this study have potential to be applied in elucidating complex heterogeneous catalysis mechanisms such as methanol-to-olefin, selective oxidations and hydrogenation reactions.

## Methods

**Synchrotron experiments.** The py-iPEPICO experiments were carried out at the VUV beamline of the Swiss Light Source[51,52]. In brief, a 150 lines per mm grating was used with a 200 μm exit slit to achieve a resolving power of around 1,500. Higher order radiation was suppressed by a rare gas filter filled with Ne or a Ne/Ar/ Kr mixture. The threshold photoionization matrix was recorded using the iPEPICO[51] endstation, which consists of a time-of-flight mass spectrometer for ions coupled with a velocity map imaging detector for photoelectrons. The electron hits act as start signal for the ion time-of-flight analysis in a multiple-start multiple-stop detection scheme[53]. The photon energy was scanned and the threshold, near-zero kinetic energy, electrons were selected to record the threshold photoionization matrix.

The catalytic reactor used in our experiments applies the temporal analysis of products (TAP) approach, introduced by Gleaves et al.[26] The design and performance is identical to the reactor published by Leppelt et al.[54] Proof-of-principle experiments and a drawing of the experimental design are presented in Supplementary Methods and Supplementary Fig. 1. The reactor source is directly connected to the spectrometer chamber of the iPEPICO endstation. Argon was saturated with guaiacol at 25 °C and a total pressure of 400 mbar vapour pressure, corresponding to a dilution of around 1:2,400. The average argon pressure in the reactor during a pulse is 0.3 mbar, resulting in a guaiacol partial pressure of $10^{-4}$ mbar ($6 \times 10^{12}$ molecules per cm³). The high dilution suppresses gas-phase reactions, results in a short detection time of only a few milliseconds, and allows us to observe the desorbed reactive intermediates and products directly. Argon also acts as a purging gas for desorbed products. The pulsed valves are directly connected to a quartz glass reactor. The reactor can be heated up to 650 °C over a length of 3.5 cm. The reactor surface was coated by dispersing H-USY (Zeochem) in diethyl ether, flushing the quartz glass reactor with the slurry, removing the solvent and calcinating the reactor in air for 5 h at 550 °C. The catalyst coating was then removed from the non-heated part of the reactor, leaving between 0.5 and 3 mg catalyst on the surface, depending on the layer thickness. After these procedures, the loaded reactor was connected to the pulsed valves and pumped by a roughing pump before the gate valve to the spectrometer chamber was opened. Due to the low vapour pressure of catechol, the reactor was operated continuously and without a buffer gas in the catechol pyrolysis experiment.

**Pyrolysis-GC/MS set-up.** The catalyst-guaiacol mixture is pyrolysed in an open-ended quartz batch reactor heated resistively by a platinum coil pyrolyzer (5150, CDS Analytical) and held in place by quartz wool. A helium carrier gas stream transfers the products at 300 °C into the GC/MS system equipped with a thermal conductivity detector (TCD, Agilent 7890 A GC and Agilent 5975 MS). The GC oven was programmed to start at 40 °C and heat up to 200 °C at a heating rate of 10 °C min⁻¹ and to 270 °C at 20 °C min⁻¹. The columns are optimized for either phenol separation (HP-5MS) or for gas separation (Plot/Q and molecular sieve capillary column). Products were identified according to the NIST08 mass spectrum library and the non-condensable gases ($CO$, $CO_2$) are identified and quantified by the TCD. Generally, the lignin pyrolysis products were calibrated with standard solutions, so that isomer-specific retention times were known. For the $^{13}C$-labelled guaiacol experiments a single product analysis of the mass spectra was performed. H-USY catalyst was used in fourfold excess to guaiacol. The heating rate of the pyrolysis was 20 °C ms⁻¹ and the pyrolysis time 20 s. Notwithstanding its susceptibility to coking with respect to HZSM-5, the zeolite H-USY was chosen as catalyst, because of its high activity and pore size, 7.4 Å, large enough to accommodate reactants and pyrolysis products based on their critical kinetic diameter[17,55]. Reactive and stable intermediates such as cyclopentadiene, fulvene and fulvenone could not be detected in the GC/MS set-up in the presence on H-USY. Despite varying several parameters such as lowering the initial oven temperature or increasing the flow rates, we could not observe 5-membered ring species. Pure cyclopentadiene on H-USY for instance was detectable using the py-GC/MS set-up along with thermodynamic reaction products such as BTX, indane and naphthalene. However, in the presence of guaiacol (4:1 with cyclopentadiene) no cyclopentadiene was desorbed from the catalyst, probably due to the high reactivity of species being formed after guaiacol decomposition (Supplementary Note 2 for more details).

**Computational.** The Gaussian 09 suite of programs was used to calculate ionization energies and reaction enthalpies, using CBS-QB3 level of theory[44]. We have used eZspectrum with the CBS-QB3 optimized geometries and Hessian matrices to calculate the Franck–Condon factors and simulate the experimental ms-TPES[56].

**Data availability.** The data that support the findings of this study are available from the corresponding author upon reasonable request.

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

## Acknowledgements

S. Sonderegger and P.H. designed the reactor used in the py-iPEPICO set-up. P.H. acknowledges useful discussions on the TAP design with Dr K. Morgan, Professor A. Goguet (Queen's University Belfast), Professor R. J. Behm (University of Ulm) and Professor J. Sa (University of Uppsala). The Swiss Federal Office of Energy (SFOE) under contract SI/501269-01 supports P.H., A.B. and T.G. We would like to thank P. Ascher for technical support and Dr Z. Ma for his input on the hydrogen transfer reactions. V.B.F.C. and J.A.v.B. kindly acknowledge funding by the Swiss National Science Foundation (NRP66, no. 406640- 136892).

## Authors contributions

P.H., V.B.F.C. and A.B. wrote the manuscript. P.H. and J.A.v.B. had the idea to perform this study. The py-iPEPICO experiments and analysis were performed by P.H., with contributions from A.B., V.B.F.C. and T.G. The py-GC/MS experiments were performed by V.B.F.C. with contributions from P.H. The data were discussed among all authors. All authors read and commented on the manuscript.

## Additional information

**Competing interests:** The authors declare no competing financial interests.

