## [Peer Review File · Nature Communications]

Reviewers' comments:

Reviewer #1 (Remarks to the Author):

The authors implemented an original experimental method to study the intermediates to shed light on the catalytic pyrolysis mechanism of guaiacol. The novelty of the approach lies in the introduction of photoelectron photoion coincidence (PEPICO) spectroscopy with synchrotron radiation to allow for isomer-selective detection of reactive intermediates. Pyrolysis is an important process and a molecular scale understanding is currently lacking. In this regard, this work makes an important step forward. The paper is publishable in Nat Comm upon revising the following:

1. The authors should replace the term intermediate with stable intermediate to avoid confusion since radicals are not detected herein.
2. In Lines 116-118: the authors claimed that the transient species did not survive on the py-GCMS interface. If there's a need, the authors could try to modify the analytical method (e.g. lowering the initial oven temperature by using liquid Nitrogen) and see if these species could be detected.
3. The peaks in the py-GCMS chromatogram were identified using the library search. However, the isomers could give equally good matches. Have the authors run any pure compounds as standards to identify the isomers more rigorously?
4. In Table 1 in supplemental information, trimethylbenzene was not detected in Py-GCMS while lighter species such as benzene and xylene could be detected. Have the authors thought about why this is the case? Is it due to the analytical methods used? The last column of the table is confusing; some of the species were identified in this work but the last column shows literature references. Are these species identified in this work only, in the literature only, or in both?
5. In the study using Na-USY catalyst, only the Py-iPEPICO setup was used. Since two set ups don't detect the same species (neither the same type nor quantity), the authors should use the Na-USY catalyst for both setups just like the work using the H-USY catalyst. Have the authors considered plotting the mass spectra from the H-USY and Na-USY experiments together?

Some mechanisms were proposed without having strong evidences:

1. In Figure 3, ¹³C labeled compounds need to be explicitly marked where they are labeled. Chemical structures should be accompanied with chemical names. In the text, the chemical names should be accompanied with their numbers. The numbering of the compounds are scattered and therefore hard to follow. It would be nice to re-number the compounds according to the reaction cascade.
2. Line 198: what is the evidence that isomerization occurs during transfer line in py-GC/MS method? Could labeled benzene form via a different pathway?
3. Line 218: What are the other radical besides methyl radicals? If they cannot be detected, what is the experimental evidence for their strong adsorption on the catalyst surface?
4. Section "Suppressing transmethylation": the authors stated that they could only detect the methyl radical. However, a radical mechanism was proposed to explain the formation of cyclopentadiene, fulvenone and phenol. Could the authors provide more basis for the proposed mechanism?

Some typos were found in the manuscript:

1. In Line 54, lignin is the third major component in biomass.
2. In Line 72, "lead" should be "led".

Reviewer #2 (Remarks to the Author):

The paper aims to elucidate the reaction pathways of pyrolysis via different methods, introducing a new method for detection of catalytically active species, the photoelectron photoion coincidence

(PEPICO) spectroscopy. This would per se be an interesting topic, as it would widen the scope of analytical methods to detect short-lived intermediates.

The current manuscript, however, falls short of convincingly showing the benefit of the detection for the analysis of the catalyzed pyrolysis pathway. This is not necessarily the fault of the authors, but maybe also due to the very elusive nature of the complex interplay between gas phase and surface catalyzed reactions. The authors do not provide sufficient evidence, which steps are gas phase, and which are surface catalyzed, leading to a scheme for guaiacol pyrolysis, which in essence adds to speculations.

I think at present the manuscript is premature from the point of bringing new chemistry to understand the catalyzed transformations targeted. It could be worth publication at a later point, once the manuscript is shortened in the introduction (overall it reads more like an activity report than a paper) and carefully addresses critical points.

1. The discussion of the reaction mechanism suffers from the lack of definition of the residence time of the reactants in the zeolite pore, and their distribution. For TAP like measurements, this requires understanding and modeling of the individual distributions. It is certain that the residence times are much longer than the higher dynamic residence time suggested.

2. Given the expected longer residence time in zeolite pores, one may question the relevance of radical species detected, as well as the central nature of the proposed five membered ring oxygenates in the zeolite pores. As the authors point out, none of these compounds were found by GC MS.

3. In the proposed mechanism, the majority of steps are speculations, rather than based on proven intermediates. This is to a large extent unavoidable, but for several elementary steps alternative reactions would be feasible (decarbonylation vs. decarboxylation for example). One may question then, what additional information the scheme brings to the state of knowledge (also in part published by some of the authors)

4. To make the problem nearly untraceable is the postulation of a carbon pool on the surface. This eliminates the possibility of a mass balance, leaving only room speculations, as every reaction can be explained by the participation of the carbon pool.

These four points are not exhaustive for the problems the manuscript; this should give the authors, however, the possibility to restructure the manuscript. If the main purpose is to show the validity of the method to trace short-lived intermediates, one may choose, preferably a simpler reaction, one which would also allow to define the differences between intra-porous and extra-porous reaction steps.

Reviewer #3 (Remarks to the Author):

The manuscript discusses the use of photoelectron photoion coincidence (PEPICO) spectroscopy to examine the catalytic conversion of guaiacol with a H-USY zeolite. Overall, I find the work to be interesting and a useful contribution to the technical literature as PEPICO appears to be able to provide some interesting information. However, I think the manuscript needs to consider several points as the authors have overstated several points:

1. The manuscript specifically looks at the gas phase conversion of guaiacol over a zeolitic catalyst. It is not at all clear why this work is being claimed as the "The Case of Catalytic Fast Pyrolysis." Yes, guaiacol is commonly used as a model compound representing a molecule generated by the pyrolysis of lignin/biomass. However, there is no catalytic fast pyrolysis in this work. They have merely volatilized guaiacol and exposed it to a catalyst. It is fine to claim that it is worthwhile to examine this model compound that is generated in fast pyrolysis.

2. The PEPICO analysis requires the reaction to be performed under vacuum. Operating the reaction under vacuum can modify the chemistry of the reaction as the concentration and types of adsorbed species will be different than under real reaction conditions. The implications of this difference would become even more extreme in real CFP in which many more species are present in the system. Again, I think it is inappropriate to make as strong of a link to the catalytic mechanisms of H-USY in CFP as the authors attempt to do in the manuscript.

The work is worthy of publication but the broad claims of utility of the approach in understanding CFP have not been demonstrated in the results and need to be modified accordingly.

Point-by-point reply: NCOMMS-17-00947

Dear Reviewers,

Please find attached our revised manuscript entitled: "Understanding Reaction Mechanisms in Heterogeneously Catalyzed Reactions: Disentangling Crucial Reactions in Catalytic Fast Pyrolysis".

We are very thankful for your constructive remarks and suggested additional experiments. We think that the revised paper benefited greatly from the comments, which we would like to address in the following:

Reviewer #1 (Remarks to the Author):

The authors implemented an original experimental method to study the intermediates to shed light on the catalytic pyrolysis mechanism of guaiacol. The novelty of the approach lies in the introduction of photoelectron photoion coincidence (PEPICO) spectroscopy with synchrotron radiation to allow for isomer-selective detection of reactive intermediates. Pyrolysis is an important process and a molecular scale understanding is currently lacking. In this regard, this work makes an important step forward. The paper is publishable in Nat Comm upon revising the following:

1. The authors should replace the term intermediate with stable intermediate to avoid confusion since radicals are not detected herein.

Answer: We thank the referee for the constructive comments on our manuscript. We decided to specify stable and reactive intermediates now as follows: Methyl radicals, fulvenone and methyl-fulvenone are termed reactive intermediates, while cyclopentadiene, fulvene and methyl cyclopentadienes are stable intermediates. We specified the stable and reactive intermediates through the whole manuscript.

2. In Lines 116-118: the authors claimed that the transient species did not survive on the py-GCMS interface. If there's a need, the authors could try to modify the analytical method (e.g. lowering the initial oven temperature by using liquid Nitrogen) and see if these species could be detected.

Answer: We know from previous studies (J. Phys. Chem. B 118, 8524-8531 (2014)) that five-membered ring species, such as cyclopentadienone and cyclopentadiene are not stable at ambient pressure pyrolysis conditions, probably due to reactions on the surface and/or in the gas phase. Trace amounts of five-ring species could only be observed when the reaction temperature was increased to 800°C and conditions were optimized. However, the relatively long residence time in the product separation column (several minutes in contrast to a few μ -seconds in py-PEPICO) already makes it almost impossible to detect labile compounds. However, the labelling experiments with ^{13}C guaiacol confirmed that the observed catalytic reactions are the same in both setups. As suggested by the referee, additional py-GC/MS experiments were carried out on the H-USY/guaiacol system to test the

absence of stable intermediates. We have used 0.8, 1.1 and 4.0 sccm flow rates on the column of the GC/MS system. In addition, the transfer line was held at 175, 250 and 325 °C, however it was never possible to detect any transient species using py-GC/MS and the selectivity and conversion was identical. Furthermore, we have introduced freshly prepared cyclopentadiene (CP) into the py-GC/MS system. Indeed, the CP survived the whole detection process and no reaction products could be observed. Upon using H-USY most of the CP stays on the catalyst, the conversion was much smaller compared to H-USY and guaiacol and only BTX have been observed as the thermodynamically most stable products. The batch-type reactor also turned brownish or black indicating coking, depending on the amount of CP introduced on the H-USY. When introducing a mixture of guaiacol and cyclopentadiene (4:1) on H-USY, cyclopentadiene completely evades detection, since it gets depleted by reaction with products and intermediates from guaiacol CFP. This observation proves that CP is stable enough to be detected using GC/MS, however, as soon as other reactants are present in the reaction mixture CP is faster consumed than desorbed from the catalyst surface. To stress this, we have added a new section in the supporting information and methods section of the paper.

3. The peaks in the py-GCMS chromatogram were identified using the library search. However, the isomers could give equally good matches. Have the authors run any pure compounds as standards to identify the isomers more rigorously?

Answer: Generally, the lignin pyrolysis products were calibrated with standard solutions, so that isomer specific retention times were known. Additionally, the analysis of the ¹³C-labeled guaiacol data was not possible with simple comparison to the NIST database, and every single product required careful analysis and identification. Specific isomers, such as methyl-cyclopentadiene (m/z=80) were not observed with py-GC/MS and can be excluded due to different molecular weight. Fulvene and fulvenone are unfortunately not stable enough for standard analysis, otherwise we would have tested for these molecules as well. A paragraph was added to the methods section to provide information on the calibrants.

4. In Table 1 in supplemental information, trimethylbenzene was not detected in Py-GCMS while lighter species such as benzene and xylene could be detected. Have the authors thought about why this is the case? Is it due to the analytical methods used? The last column of the table is confusing; some of the species were identified in this work but the last column shows literature references. Are these species identified in this work only, in the literature only, or in both?

Answer: We did not observe trimethylbenzenes in the py-iPEPICO setup, but up to 7% were present in the py-GC/MS setup. The lack of trimethylbenzenes in the py-iPEPICO setup can be explained by concentration effects. The py-iPEPICO approach uses much smaller reactant densities on the surface, therefore the triply methylated products are mostly absent.

Indeed, the references in table 1 of the SI are misleading. All compounds were observed in this study; however, the references are simply the literature ionization energies or photoelectron spectra, which we have used to identify the species in the py-iPEPICO setup. We have corrected this in the revised version of the SI, page 9, table 1.

5. In the study using Na-USY catalyst, only the Py-iPEPICO setup was used. Since two set ups don't detect the same species (neither the same type nor quantity), the authors should use the Na-USY catalyst for both setups just like the work using the H-USY catalyst. Have the authors considered plotting the mass spectra from the H-USY and Na-USY experiments together?

Answer: Thanks for bringing up this point. Na-USY was investigated in the py-GC/MS setup as well. We found almost no reactivity at 400°C comparable with the PEPICO data. At 500 °C the conversion was a little bit higher but still smaller compared to H-USY. Due to the lack of conversion at similar reaction conditions with Na-USY in the py-GC/MS setup we did not consider a comparable plot. Only at higher temperature transalkylation products could be detected along with catechol and other products from pure bimolecular chemistry such as diphenylether and hydroxybenzaldehyde, which are also observed in absence of a catalyst (see J. Phys. Chem. B 118, 8524-8531 (2014)). We have summarized this in the manuscript as well on p. 11 (last paragraph).

Some mechanisms were proposed without having strong evidences:

1. In Figure 3, ^{13}C labeled compounds need to be explicitly marked where they are labeled. Chemical structures should be accompanied with chemical names. In the text, the chemical names should be accompanied with their numbers. The numbering of the compounds is scattered and therefore hard to follow. It would be nice to re-number the compounds according to the reaction cascade.

Answer: Thanks. We have labeled all the products with an asterisk, which appear ^{13}C labeled and renumbered Figure 3 according to the reaction cascade.

2. Line 198: what is the evidence that isomerization occurs during transfer line in py-GC/MS method? Could labeled benzene form via a different pathway?

Answer: We cannot tell whether the isomerization of fulvene to benzene happens in the micro-reactor or it occurs in the transfer line or even during the GC separation process. We have now modified the paragraph on p. 10 correspondingly. In addition, the total residence time in the py-GC/MS setup is in the order of minutes, which is orders of magnitude longer compared to the rapid probing in the py-iPEPICO setup.

Indeed, the fact that benzene is only labelled once under these conditions, together with the detection of cyclopentadiene, ^{13}C -methyl-cyclopentadiene and ^{13}C -fulvene proves the existence of the reaction pathway to produce benzene. The alternative pathway would be the complete destruction and reconstruction of the aromatic moiety. However, since benzene was always at most singly ^{13}C labeled, we can rule this out. In addition, phenol, catechol and fulvenone always appeared unlabeled. We have added a few sentences clarifying these issues on p.10.

3. Line 218: What are the other radical besides methyl radicals? If they cannot be detected, what is the experimental evidence for their strong adsorption on the catalyst surface?

Answer: We have only detected the methyl radical intermediate using Na-USY, which is the result of homolytic bond cleavage of the O-CH₃ bond at the methoxy group. The fact that the hydroxyphenoxy counter radical has not been observed shows that it is stronger bound to the catalyst surface than the barrier to decomposition to form cyclopentadienone (c-C₅H₄=O), which is ultimately desorbed and identified in the py-iPEPICO setup. Furthermore, we observe strong guaiacol adsorption on the catalyst, evidenced by the drop in guaiacol signal by several orders of magnitude upon introduction of the acidic zeolite catalyst. In Figure S4 (and Fig. 4 in the case of catechol) you can observe the increase of guaiacol signal at a time-on-stream > 200 min, which simply occurs due to deactivation of the catalyst, i.e., blockage and inactivity of the adsorbing acid sites. In the work of Bährle et al. zeolites showed a radical concentration enhancing effect, as measured by EPR spectroscopy, corresponding to the acidity of the zeolites (ChemSusChem, 2016, 9, 2397) during lignin catalytic pyrolysis. This proves that radicals remain within the zeolite during catalytic pyrolysis or are chemisorbed on the surface. These phenomena are due to adsorption onto and stabilization by the surface (especially due to the low pressure). We added a paragraph on page 11 which emphasizes these arguments.

4. Section "Suppressing transmethylation": the authors stated that they could only detect the methyl radical. However, a radical mechanism was proposed to explain the formation of cyclopentadiene, fulvenone and phenol. Could the authors provide more basis for the proposed mechanism?

Answer: A correlation between CO₂ and cyclopentadiene (cp) formation is found in the time-on-stream curve (Fig 4c). While the mechanism is dominated by hydration of fulvenone at low coke concentrations, leading to the carboxylic acid (c-C₅H₅-CO₂H), which decarboxylates to yield cp and CO₂. At later times (coke builds up), the CO₂ signal decreases while the cp signal levels off, speaking for a second formation pathway of cp, which is probably initiated by hydrogen transfer reactions from condensed polyaromatics. Although this process of hydrogen addition to the fulvenone species may proceed in a concerted fashion, the radical intermediates after formal hydrogen atom addition may explain the branching into cyclopentadiene and phenol. We know from gas phase experiments (Phys.Chem.Chem.Phys., 2015, 17, 30076), that phenoxy radicals decompose to yield cyclopentadienyl radicals. In this context, phenoxy radicals may either be stabilized on the surface to

yield phenol or decarbonylate to form cyclopentadiene, leading to a branching. As pointed out in question 3, radical species were observed on the zeolite surface using EPR techniques. Moreover, those phenoxy or cyclopentadienyl species may be strongly bound to the catalyst due to chemisorption and are thus rather immobile and evade detection even in the py-iPEPICO setup. These species will likely add hydrogen from condensed polyaromatics in exothermic reactions and finally get desorbed. It can be inferred that upon hydrogen atom transfer radical intermediates are formed, which contribute to the spin density as measured by EPR spectroscopy (see Question 3). We have modified p.14 of the manuscript to emphasize that the branching into cyclopentadiene and phenol is well explained by radical intermediates, however the whole reaction may proceed concerted.

Some typos were found in the manuscript:

1. In Line 54, lignin is the third major component in biomass.

2. In Line 72, “lead” should be “led”.

Answer: The typos have been corrected accordingly.

Reviewer #2 (Remarks to the Author):

The paper aims to elucidate the reaction pathways of pyrolysis via different methods, introducing a new method for detection of catalytically active species, the photoelectron photoion coincidence (PEPICO) spectroscopy. This would per se be an interesting topic, as it would widen the scope of analytical methods to detect short-lived intermediates. The current manuscript, however, falls short of convincingly showing the benefit of the detection for the analysis of the catalyzed pyrolysis pathway. This is not necessarily the fault of the authors, but maybe also due to the very elusive nature of the complex interplay between gas phase in surface catalyzed reactions. The authors do not provide sufficient evidence, which steps are gas phase, and which are surface catalyzed, leading to a scheme for guaiacol pyrolysis, which in essence adds to speculations.

Answer: We would like to emphasize in passing the proven utility of py-GC/MS setup in catalysis, although it is not able to distinguish between reactions on the catalyst surface and in the gas phase. On the other hand, we agree with the referee that this is a crucial issue, and our py-PEPICO experiments were motivated precisely by the virtually complete suppression of gas phase reactions due to the high dilution of guaiacol in argon (1:2400) and the low average pressure in the reactor (0.3 mbar total pressure). We have described this now in more detail in the “Bridging the pressure gap” section of the manuscript (p.4 and p.5). Furthermore, we have studied bimolecular gas phase reactions guaiacol and other phenoxy species (J. Phys Chem B, 2014, 118(29), 8524), and the products, e.g., cyclopentadienon or 2-hydroxy-benzaldehyde were not observed in py-PEPICO. Lastly, the reaction temperature is significantly lower than required for initiating bimolecular gas phase chemistry, and the time-on-stream curves also show features only consistent with surface chemistry. This shows conclusively that py-PEPICO is indicative of surface-based reactions and gas phase chemistry does not interfere with any of our results.

On page 16 (conclusion) we added the following sentences to emphasize what are likely the surface reactions, what the gas phase ones: “Transmethylation, hydration, dehydration reactions as well as the hydrocarbon transfer happen surface-catalyzed, as confirmed by the presence of the same products in both setups. Furthermore, typical gas phase reaction products of guaiacol pyrolysis such as diphenylether and hydroxybenzaldehyde were absent in the presence of the H-USY catalyst. Reactive intermediates, such as methyl radicals and the fulvenone ketene could only be detected in the py-iPEPICO setup speaking for their rapid quenching in the ambient pressure py-GC/MS experiments in the gasphase.

I think at present the manuscript is premature from the point of bringing new chemistry to understand the catalyzed transformations targeted. It could be worth publication at a later point, once the manuscript is shortened in the introduction (overall it reads more like an activity report than a paper) and carefully addresses critical points.

Answer: We have shortened the introduction now and took care of the critical points:

1. The discussion of the reaction mechanism suffers from the lack of definition of the residence time of the reactants in the zeolite pore, and their distribution. For TAP like measurements, this requires understanding and modeling of the individual distributions. It is certain that the residence times are much longer than the higher dynamic residence time suggested.

Answer: The referee is right about the residence time on the surface. As discussed in the original supporting information, no overall pulse structure of guaiacol travelling through the catalyst-coated reactor could be observed, speaking for residence times on the surface significantly longer than the gas pulse duration. We have added a sentence in the supporting information (p 3). The time-on-stream curves in Figure S4 provide some insights and significant desorption can only be observed several minutes after starting the experiment, resulting in residence times of several minutes on the surface. Modeling the individual distributions is beyond the scope (detecting reactive intermediates) of this manuscript and probably requires a clearly defined system of model compound and catalyst and would not lead to additional insights. It is one of the important observations that adsorption in the zeolite pores leads to long residence time during CFP.

2. Given the expected longer residence time in zeolite pores, one may question the relevance of radical species detected, as well as the central nature of the proposed five membered ring oxygenates in the zeolite pores. As the authors point out, none of these compounds were found by GC MS.

Answer: The referee is right to point out radicals and fulvenones were not observed in the GC/MS. The GC/MS setup has a higher molecular concentration (partial pressure) and therefore more initial loading of the reactive compounds. As mentioned in the manuscript (p.7, first paragraph) it is known that fulvenone is quickly (probably within nanoseconds) quenched by water to form a carboxylic acid and may dimerize already below room temperature. In addition, much higher concentrations of guaiacol in the batch-type reactor setup can also quickly lead to stable reaction products and reactive molecules evade detection in GC/MS because of the long detection time, anyway. Again, this comment accurately mirrors one of the central points of the present work: to show that the underlying catalytic mechanism is the same in the GC/MS, yet quickly detect intermediates that may desorb from the surface by suppressing of gas phase chemistry, and use this to gain further insights into the catalytic process.

3. In the proposed mechanism, the majority of steps are speculations, rather than based on proven intermediates. This is to a large extent unavoidable, but for several elementary steps alternative reactions would be feasible (decarbonylation vs. decarboxylation for example). One may question then, what additional information the scheme brings to the state of knowledge (also in part published by some of the authors).

Answer: Alternative pathways e.g. decarbonylation of fulvenone to yield C_5H_4 species may contribute to the formation of PAHs such as naphthalene. Its low concentration in the GC/MS experiment, however, refutes this mechanism. Our present scheme explains the branching of fulvenone to phenol and cyclopentadiene, which is the precursor of fulvene and benzene. To the best of our knowledge the fulvenone intermediate was not considered previously, and its central role is discussed here for the first time in the deoxygenation of catechol. We have used labelling to observe intermediates and infer missing links in Figure 3. One of the aims of this work was to find a feasible reaction pathway for the formation of benzene from lignin based phenol. The question was whether it is formed by complete fragmentation of the phenols and condensation to stable final products, BTX, or via other reactive intermediates. The ^{13}C -labelling helped to observe that direct deoxygenation may occur in this process. But also singly ^{13}C -labelled benzene was a dominant product, so that there are two parallel mechanisms.

4. To make the problem nearly untraceable is the postulation of a carbon pool on the surface. This eliminates the possibility of a mass balance, leaving only room speculations, as every reaction can be explained by the participation of the carbon pool.

Answer: We disagree with this argument. Char formation is well established in pyrolysis and also observed in our experiments. We did not merely postulate a carbon pool, but actually observed its formation. That coke is formed by polyaromatics having lower H/C ratio is also well established. Indeed, highly condensed polyaromatics were found using ^{13}C MAS solid state NMR spectroscopy (ChemCatChem 10.1002/cctc.201601674). We have added a sentence on p.14 clarifying this statement. Therefore, the availability of hydrogens from condensed polyaromatics and the detection of cyclopentadiene and phenol, provides conclusive evidence for the branching of fulvenone, also mirrored by the reactivity of phenoxy species (*Phys.Chem.Chem.Phys.*, 2015, 17, 30076), which yield cyclopentadienyl radicals and may be extrapolated from the gas phase to the catalyst surface (see also Referee 1, Answer 4). Since the term hydrocarbon pool might be misleading, due to its common use in FCC cracking, we use H-transfer reactions from highly condensed aromatics instead. We have rephrased p. 13 – 14 accordingly.

These four points are not exhaustive for the problems the manuscript; this should give the authors, however, the possibility to restructure the manuscript. If the main purpose is to show the validity of the method to trace short-lived intermediates, one may choose, preferably a simpler reaction, one which would also allow to define the differences between intra-porous and extra-porous reaction steps.

Answer: We do not share the criticism of the reviewer that we made a poor choice of reaction. The merit of the technique is illustrated best by its ability to identify complex surface chemistry reaction paths based on the experimentally observed (and, in some cases short-lived) intermediates and labelling experiments. Several referee comments in fact highlight in a critical light what we consider to be the fortes of the py-PEPICO technique, namely the suppression of gas phase chemistry and the detection of intermediates that are rapidly quenched at higher pressures or do not survive until detection in alternative analytical approaches. We have reviewed the relevant sections to emphasize these points more clearly, and hope to convey the advantages of the technique as clear positives in the revised manuscript. We also agree that a different, simpler choice of reaction would allow to define differences between intra-porous and extra-porous reaction steps. However, this was not the aim of the manuscript, and py-PEPICO may not be the best detection method for such a study. Instead, our goal was to make an important step towards understanding CFP, and we think we were able to show the potential of the py-PEPICO technique to study surface chemistry in catalysis.

Reviewer #3 (Remarks to the Author):

The manuscript discusses the use of photoelectron photoion coincidence (PEPICO) spectroscopy to examine the catalytic conversion of guaiacol with a H-USY zeolite. Overall, I find the work to be interesting and a useful contribution to the technical literature as PEPICO appears to be able to provide some interesting information. However, I think the manuscript needs to consider several points as the authors have overstated several points:

1. The manuscript specifically looks at the gas phase conversion of guaiacol over a zeolitic catalyst. It is not at all clear why this work is being claimed as the "The Case of Catalytic Fast Pyrolysis." Yes, guaiacol is commonly used as a model compound representing a molecule generated by the pyrolysis of lignin/biomass. However, there is no catalytic fast pyrolysis in this work. They have merely volatilized guaiacol and exposed it to a catalyst. It is fine to claim that it is worthwhile to examine this model compound that is generated in fast pyrolysis.

Answer:

We understand the concern whether this reaction is relevant for the catalytic fast pyrolysis of lignin. During catalytic fast pyrolysis of lignin, the catalytic reaction does not occur directly between the bulk lignin polymer and the catalyst, but lignin has to depolymerize first and the products then diffuse in the zeolites. Literature and our previous studies suggest that single phenolic compounds are generated and are consequently effectively converted over the zeolite. There is an increasing number of studies

which separate these two steps, because lignin depolymerization may require different conditions (temperature) than the catalytic conversion of its products. We have revised our choice of words to reflect this and modified the title to “Understanding Reaction Mechanisms in Heterogeneously Catalyzed Reactions: Disentangling Crucial Reactions in Catalytic Fast Pyrolysis”.

2. The PEPICO analysis requires the reaction to be performed under vacuum. Operating the reaction under vacuum can modify the chemistry of the reaction as the concentration and types of adsorbed species will be different than under real reaction conditions. The implications of this difference would become even more extreme in real CFP in which many more species are present in the system. Again, I think it is inappropriate to make as strong of a link to the catalytic mechanisms of H-USY in CFP as the authors attempt to do in the manuscript.

Answer: In order to ensure the validity of our reaction mechanism in the py-iPEPICO setup, we compare the detected products and stable intermediates with the more realistic ambient pressure py-GC/MS results. As Figure 1 shows and is discussed in the paper, the final products of catalytic conversion are by large the same and differences, *e.g.*, in the methylation degree can easily be explained by the higher pressures/concentrations in py-GC/MS. Therefore, we are in fact confident that the underlying surface chemistry is not significantly affected by the low pressure environment in the py-iPEPICO setup.

We also believe that guaiacol represents both common lignin functionalities, such as a methoxy group together with a hydroxyl group. Other common lignin depolymerization products with multiple methoxy groups or vinyl substituents are much more reactive. Establishing the guaiacol conversion mechanism is a necessary first step in deducing the behavior of more complex compounds, which we also intend to address in the future. Nevertheless, we have softened our statement in the manuscript: “We infer that this intermediate (fulvenone) may also play a crucial role in deoxygenation of lignin in CFP.” (see conclusion)

The work is worthy of publication but the broad claims of utility of the approach in understanding CFP have not been demonstrated in the results and need to be modified accordingly.

Answer: Some of the statements in the original submission may have been too broadly phrased. We have made these more specific now, as the referee rightly suggests that they may or may not apply to catalytic fast pyrolysis of lignin in general. However, we still believe that we successfully identified major intermediates and, thus, reaction pathways in the deoxygenation of guaiacol into phenol and benzene, which illustrates to capability of py-iPEPICO to interrogate surface chemistry in a unique way by isomer selectively identifying minute quantities of desorbed reaction intermediates not accessible before.

REVIEWERS' COMMENTS:

Reviewer #1 (Remarks to the Author):

The authors have satisfactorily addressed the comments from the reviewers to clear up confusion regarding the text, formatting, and experimental methods. Specifics are appended. However, the proposed mechanism remains largely speculated. Although the authors provided findings from literature to support their claims, e.g., the strong chemisorption on the catalyst surfaces of the radical intermediates, the rationalization for the proposed mechanism is convoluted and needs to be clearer and more concise. I propose the paper gets revised.

1. The title has been modified to better reflect the contents of the paper, and is a better title than the previous one containing lignin CFP.
2. The typos in the text have been corrected.
3. The Introduction has become more concise.
4. Stable reaction intermediates and short-lived radicals have been defined and distinguished.
5. Scheme showing proposed reaction pathway (Fig. 3) has been clearly color-coded and labeled with species names, numbers, and reaction mechanisms.
6. Reaction using Na-USY catalyst in the Py-GCMS setup has been summarized to parallel the reaction using the same catalyst in the Py-iPEPICO setup.
7. Table 1 in supplemental information has been revised to clear up the previous confusion about whether or not the species given were detected by the authors or in the past literature.
8. Methods used for identifying the species using GC-MS have been clearly specified.

Reviewer #3 (Remarks to the Author):

The authors have appropriately addressed the reviewer's comments and have made key modification to the manuscript. Therefore, the manuscript is worthy of acceptance.

Point-by-point response for the manuscript (NCOMMS-17-00947A): „Understanding the mechanisms of catalytic fast pyrolysis by unveiling reactive intermediates in heterogeneous catalysis”

Reviewer #1 (Remarks to the Author):

The authors have satisfactorily addressed the comments from the reviewers to clear up confusion regarding the text, formatting, and experimental methods. Specifics are appended. However, the proposed mechanism remains largely speculated. Although the authors provided findings from literature to support their claims, e.g., the strong chemisorption on the catalyst surfaces of the radical intermediates, the rationalization for the proposed mechanism is convoluted and needs to be clearer and more concise. I propose the paper gets revised.

A: We are grateful for the helpful comments and remarks and feel that the manuscript benefitted greatly. The part about the strong chemisorption which explains the absence of reactive intermediates was rephrased according to the reviewer's comment (see p. 11).

- 1. The title has been modified to better reflect the contents of the paper, and is a better title than the previous one containing lignin CFP.***
- 2. The typos in the text have been corrected.***
- 3. The Introduction has become more concise.***
- 4. Stable reaction intermediates and short-lived radicals have been defined and distinguished.***
- 5. Scheme showing proposed reaction pathway (Fig. 3) has been clearly color-coded and labeled with species names, numbers, and reaction mechanisms.***
- 6. Reaction using Na-USY catalyst in the Py-GCMS setup has been summarized to parallel the reaction using the same catalyst in the Py-iPEPICO setup.***
- 7. Table 1 in supplemental information has been revised to clear up the previous confusion about whether or not the species given were detected by the authors or in the past literature.***
- 8. Methods used for identifying the species using GC-MS have been clearly specified.***

A: No additional actions were needed.

Reviewer #3 (Remarks to the Author):

The authors have appropriately addressed the reviewer's comments and have made key modification to the manuscript. Therefore, the manuscript is worthy of acceptance.

A: We thank the referee for taking time to look a second time on the manuscript. No additional actions were needed.